# Vitamin D and the risk of dystocia: A case-control study

**Christine Rohr Thomsen**[1]*, **Ioanna Milidou**[2], **Lone Hvidman**[1], **Mohammed Rohi Khalil**[3], **Lars Rejnmark**[4], **Niels Uldbjerg**[1]

1 Department of Obstetrics and Gynecology, Aarhus University Hospital, Aarhus, Denmark, 2 Department of Pediatrics, Hospitals of West Jutland, Herning, Denmark, 3 Department of Obstetrics and Gynecology, Center Hospital Lillebaelt, Kolding, Denmark, 4 Department of Internal Medicine and Endocrinology, Aarhus University Hospital, Aarhus, Denmark

* christinerohrthomsen@gmail.com

## Abstract

### Background

Dystocia is one of the most common causes of cesarean section in nulliparous women. Studies have described the presence of vitamin D receptors in the myometrium, but it is still unclear whether vitamin D affects the contractility of the smooth muscles. We therefore aimed to determine the association between the vitamin D serum level at labor and the risk of dystocia.

### Method

We conducted a case-control study between January 2012 and June 2017. Cases were primiparous women, with spontaneous onset of labor, who gave birth by cesarean section due to dystocia. Controls were primiparous women with a spontaneous vaginal delivery. We included 60 women (30 cases and 30 controls) in the analysis. The differences between cases and controls were assessed using chi-squared test for categorical variables and two-sample t-test or unequal t-test for continuous variables, as appropriate, after evaluation of whether they followed the normal distributions.

### Results

The mean serum 25-hydroxyvitamin D concentrations were 53.1nmol/l (95%CI; 45.2 to 60.9) among cases and 69.9nmol/l (95%CI; 57.5 to 82.4) among controls ($P$ = 0.02). The mean plasma parathyroid hormone levels were 2.25 pmol/l and 2.38, respectively ($P$ = 0.57). Even though 78% of all women reported taking a minimum of 10μg/day of vitamin D throughout pregnancy, 43% had vitamin D insufficiency, defined as serum 25-hydroxyvitamin D levels below 50nmol/l.

### Conclusions

In a Danish group of women having a cesarean section due to dystocia, we found decreased vitamin D levels.

**Data Availability Statement:** The data underlying the results presented in the study are submitted to the The Danish National Archive (https://www.sa.dk/en), where it will be available upon request. Researchers can, with no restrictions, get access,

to data in an anonymized form by writing an email to mailboxDDA@sa.dk. If a researcher wishes to get access to data containing personal data, an approval from The Danish Data Protection Agency is needed. All information can be found at; https://www.sa.dk/en/research-researchers-research-service-the-danish-national-archives/use-the-danish-national-archives-survey-data/.

**Funding:** This project is a result of a grant from Grosserer L.F. Foghts Foundation; www.foghtsfond.dk, no grant number. Direktør Jacob Madsen & Hustru Olga Madsens Foundation, www.jacob-og-olga-madsens-fond.dk, grant number 4947. And Aase og Ejnar Danielsens Foundation; www.danielsensfond.dk, grant number 10-000580. All three grants were received by CRT. The Aarhus University Research Foundation has supplied a scholarship to Christine Rohr Thomsen; grant number 2011-21872-88. The funders had no role in study design, data collection and analysis, decision to publish, or preparation of the manuscript.

**Competing interests:** The authors have declared that no competing interests exist.

**Abbreviations:** S-25OHD, serum 25-hydroxyvitamin D; P-PTH, plasma parathyroid hormone; PTH, parathyroid hormone.

## Introduction

Vitamin D insufficiency during pregnancy is a worldwide public health problem [1, 2]. Studies have reported a prevalence ranging from 18–84% [1] depending on the county of residence and local clothing customs, with the Nordic countries having a high prevalence. In a Danish study, one third of all pregnant women had a serum 25-hydroxyvitamin D (S-25OHD) below 50nmol/l [3], which is the cut-off level for insufficiency according to the Danish National Board of Health [4]. This insufficiency is presumably due to the lack of dermal vitamin D production from October to April, resulting from the insignificant UVB radiation in the sunlight during this period. Maternal vitamin D insufficiency may be associated with several adverse pregnancy complications, including preeclampsia [5–7] and gestational diabetes mellitus (GDM) [8].

It is still unknown whether vitamin D affects the contractility of the myometrium. However, at least two pathways are likely: one involving the intracellular vitamin D receptor (VDR) and another related to changes in the calcium metabolism [9]. The VDR is a transcription factor that mediates most of the effects of vitamin D through regulation of the expression of several genes [10]. The VDR is present in non-vascular smooth muscle [11] including the myometrium [12]. A low concentration of extracellular calcium or inhibition of the entry of the calcium ion into the smooth muscle cell reduces the sensitivity to oxytocin and thereby affects the contractility of the myometrium [13]. Thus, the concentration of calcium, largely determined by the level of vitamin D, is of importance for labor contractions [9, 13].

As dystocia accounts directly or indirectly for 30–60% of all cesarean deliveries [14–16], we aimed to determine the relation between the vitamin D level at labor and the risk of having an acute cesarean section due to dystocia. We thus conducted a case-control study among a Danish population characterized by being homogeneous in terms of socioeconomic and nutritional status.

## Material and methods

We included both cases and controls from January 2012 until June 2017 at the maternity wards at Aarhus University Hospital and Center Hospital Lillebaelt, Kolding. Due to other tasks as well as maternity leave and vacation held by the research group, there were weeks without inclusions, which may have elongated the inclusion period.

Inclusion criteria for the cases were primiparous women with spontaneous onset of labor and cephalic presentation of the fetus, who gave birth by cesarean section due to dystocia. Dystocia was defined in accordance with the local guideline (Table 1). The inclusion criteria for the controls were primiparous women with spontaneous, vaginal birth. For each case, a control was included within two weeks to ensure similar sunlight exposure. In order to avoid selection bias, the controls were randomly chosen from a numbered list of those eligible daily. All cases as well as controls were included within 48 hours after giving birth. The medical records were subsequently reviewed for all cases as well as controls to ensure they fulfilled the in- and exclusion criteria.

We excluded women as follows: below 20 or above 40 years of age; with a pre-pregnancy Body Mass Index (BMI) over 30 kg/m$^2$; of non-Caucasian origin; with complications of pregnancy (preeclampsia, gestational diabetes mellitus or hypertension); giving birth before 37 or after 41 completed gestational weeks; with induction of labor (including PROM not followed by contractions); any mal presentation of the fetus like breech-, shoulder- and face presentation,; parathyroid-, renal-, liver- and, gastrointestinal diseases; diseases which affect the metabolism of vitamin D or calcium; and with drug abuse or consumption of more than 7 units of alcohol per week in the third trimester. In order to study women with dystocia because of reduced contractility of the myometrium, we excluded women with cephalo-pelvic

**Table 1. Definitions of stages and phases of labor and diagnostic criteria for dystocia, modified from Kjærgaard [17].**

| Stage of labor | Definition of stage and phase | Inclusion criteria (dystocia) |
|---|---|---|
| First stage | From onset of regular contractions leading to cervical dilatation to full dilatation | |
| Latent phase | Orificium < 4 cm | Cases not included in this phase. |
| Active phase | Orificium ≥ 4 cm | Progression of cervical dilatation less than 0.5 cm per hour assessed over 3–4 hours |
| Second stage | From full dilatation to delivery | |
| Descending phase | From full dilatation to strong and irresistible urge to push | More than two hours without descent |
| Expulsive phase | Strong and irresistible pushing during the major part of the contraction | More than one hour without progress |

disproportion/obstruction; obstetricians with more than 15 years of clinical experience made this distinction (NU, LH & MK).

Information about the pre-pregnancy weight, height, smoking habits, and the use of epidural analgesia during labor were obtained by reviewing the medical record. If the woman had smoked at any point during the pregnancy, she was recorded as a smoker. The last weight in pregnancy was self-reported at inclusion. Information about vitamin D supplement, if any, during the pregnancy was also acquired at inclusion by showing the woman a pictured list of all available vitamin D supplements.

## Diagnostic test

We assessed the S-25OHD and the plasma parathyroid hormone (P-PTH) concentrations by obtaining blood samples at the day of inclusion between 9 am and noon (12 am) in order to avoid any influence by the potential circadian rhythm of PTH. None of the participants was receiving an IV infusion at the time of the venipuncture. Because the serum half-life for S-25OHD is about 3 to 5 weeks, we expect that S-25OHD levels was only minimally influenced by the short fasting or changes in dietary intake, which characterizes women giving birth. Each blood sample was centrifuged at 3,000 rounds per minute for 10 min, and then EDTA plasma and serum were stored at -80˚C until analysis. After inclusion of all subjects, analyses were performed in batch. Serum levels of 25OHD were quantified by isotope dilution liquid chromatography—tandem mass spectrometry by a method adapted from Maunsell et al [18], and described elsewhere in detail [19]. The method quantifies $25OHD_2$ and $25OHD_3$, including the 3-epimer form that is not separated from $25OHD_3$. Calibrators traceable to NIST SRM 972 (ChromSystems, Gräfelfing, Germany) were used. Commutability was confirmed directly to NIST SRM 972 levels 1–4, and the sum of $25OHD_3$ and its epimer were compared. The mean coefficients of variation for $25OHD_3$ were 6.4% and 9.1% at levels 66.5 and 21.1nmol/l, respectively, and for $25OHD_2$ the coefficient of variation values were 8.8% and 9.4% at levels for 41.2 and 25.3 nmol/l, respectively [18]. P-PTH was measured using a second-generation electrochemiluminescent immunoassay (ECLIA) on an automated instrument (Cobas e601; Roche Diagnostics, GmbH, Mannheim, Germany). According to the manufacturer, the reference interval for PTH is 1.6–6.9 pmol/l. The lower limit of detection was 0.127 pmol/l, and total imprecision (CV %) was 3.3% at 3.69 pmol/l and 2.7% at 26.6 pmol/l.

## Statistical analyses

The differences between groups were assessed using chi-squared test for categorical variables according to the Danish National Board of Health's [4] predefined level of vitamin D

insufficiency, and two-sample t-test or unequal t-test for continuous variables, as appropriate, after evaluation of whether the two samples had the same standard deviation (SD). In addition, a paired t-test analysis were performed, as appropriate for matched case-control studies [20]. Furthermore, we calculated correlation between studied variables by calculation of Spearman's oh (ρ). Statistical significance was defined as a two-tailed P value < 0.05. Statistical analyses were conducted using STATA/IC 11.2.

We performed a linear regression analysis on the levels of S-25OHD and PTH according to the case and control status. We included the following predefined variables: maternal BMI at term, birth weight of the child, duration of gestation, maternal height, and the use of epidural analgesia during labor, which are known to be related to the risk of dystocia.

Finally, in order to increase the comparability of the two groups, we excluded the five cases with the highest pre-pregnancy BMI and their matched controls; the five cases with the highest BMI at term and their controls; the five cases with the highest birth weight and their matched controls; the five cases with the longest duration of pregnancy and their matched controls, as well as the five cases with the lowest maternal height and their matched controls; and repeated the analysis.

## Ethical approval

The study was conducted according to the Declaration of Helsinki II and approved by the Danish Regional Committee on Health Research Ethics (1-10-72-298-12) and the Danish Data Protection Agency (2012-58-006). Each participant gave written informed consent. All data used for this study are accessible through The Danish National Archive [21].

## Results

From January 2012 until June 2017, we included 30 cases and 30 controls. The participation rate was generally high, with less than 10% of women refusing to participate in the study. The cases had a higher BMI at term; a higher gestational age; gave birth to larger infants and more cases used epidural analgesia during labor compared to the controls (Table 2). The average time between when a case and the matched control were included in the study was 3.3 days, with a maximum of 9 days. None of the included women took any drugs affecting the vitamin D metabolism or the birth process.

Women with an acute cesarean section due to dystocia had a lower mean S-25OHD than women with a spontaneous vaginal delivery (53.1 nmol/l vs. 69.9 nmol/l, *P* = 0.02) (Table 3). This result did not change using linear regression adjusting for maternal BMI at term, birth

**Table 2. Background characteristics.**

| Variable | Cases, mean | 95% CI | Controls, mean | 95% CI | *P*-value |
|---|---|---|---|---|---|
| N | 30 | | 30 | | |
| Age at delivery (years) | 29.4 | 27.6 to 31.1 | 28.4 | 27.0 to 29.9 | 0.40 |
| Maternal height (cm) | 166.1 | 163.5 to 168.7 | 169.4 | 166.8 to 172.0 | 0.07 |
| Maternal pre-pregnancy BMI (kg/m2) | 22.6 | 21.6 to 23.6 | 21.5 | 20.7 to 22.3 | 0.09 |
| Maternal BMI at term (kg/m2) | 28.7 | 27.5 to 29.9 | 26.5 | 25.4 to 27.7 | **0.009** |
| Duration of gestation (completed gestational weeks + days) | 40+4 | 40+2 to 40+6 | 40+0 | 39+4 to 40+3 | **0.02** |
| Birth weight (gram) | 3791 | 3632 to 3950 | 3491 | 3369 to 3612 | **0.003** |
| Women smoking in the pregnancy n (%) *(chi squared)* | 2(6.7%) | NA | 3(10.0%) | NA | *0.64* |
| Use of epidural analgesia during labor n (%) *(chi squared)* | 13(43.3%) | NA | 3(10%) | NA | ***0.004*** |

**Table 3. Maternal S-25OHD and P-PTH levels after delivery.**

| Variable | Cases, mean | 95% CI | Controls, mean | 95% CI | *P*-value | RD | 95% CI | RD adjusted | 95% CI |
|---|---|---|---|---|---|---|---|---|---|
| S-25OHD (nmol/l) | 53.1 | 45.2 to 60.9 | 69.9 | 57.5 to 82.4 | **0.02** | 16.9 | 2.5 to 31.3 | 19.3 | 1.3 to 37.2 |
| P-PTH (pmol/l) | 2.25 | 1.97 to 2.53 | 2.38 | 2.03 to 2.72 | 0.57 | 0.13 | -0.31 to 0.56 | -0.01 | -0.5 to 0.5 |

weight of the child, duration of gestation, maternal height and the use of epidural analgesia during labor ($P = 0.02$). The paired t-test analysis displayed the same association ($P = 0.008$).

Similar results were obtained by repeating our analysis after excluding the five cases with the highest pre-pregnancy BMI and their matched controls; the five cases with the highest BMI at term and their controls; the five cases with the highest birth weight and their matched controls; the five cases with the longest duration of pregnancy and their matched controls, as well as the five cases with the lowest maternal height and their matched controls.

Among the 60 participants, 47 (78%) had an intake of 10µg vitamin D per day or more during their pregnancy. Nevertheless, 26 women (43% of all 60 women) had vitamin D insufficiency defined as an S-25OHD level below 50nmol/l. The mean vitamin D intake were 8.2µg among women with vitamin D insufficiency, and 10.4µg among women with sufficient vitamin D level, ($P = 0.13$). Serum 25-OHD levels correlated significantly with daily dose of vitamin D from supplements (ρ 0.264; p = 0.04).

None of the participants had secondary hyperparathyroidism (PTH $> 6.9$ pmol/l), but 20% (23% of cases vs. 17% of controls) had PTH levels below lower limits of the reference interval (Table 4).

## Discussion

This case-control study demonstrated a significant association between the S-25OHD and the risk of having an acute cesarean section due to dystocia. Furthermore, this study showed 43% of the participants to be vitamin D insufficient despite the fact that more than 78% had been taking 10µg or more of vitamin D per day during their pregnancy combined with a varied diet as recommended by the Danish Health Authorities [22].

A major strength of this study was the strict criteria defining cases with dystocia as well as the laboratory standards. However, the definition of dystocia is based on the local guideline, i.e. other definitions of dystocia could also have been used. In addition, the participants were generally healthy and homogenous in terms of socioeconomic and nutritional status. The cases and controls were matched on time of season to ensure our two groups were comparable regarding sunlight exposure and, thus, the level of dermal vitamin D synthesis. A limitation of this study was the lack of background information available on the women who refused to participate in the study (under 10%). Furthermore, in accordance with the local guideline the use of epidural analgesia was not addressed in the definition of dystocia, which might be considered as a limitation to the study. However, our results did not change when adjusting for the use of epidural analgesia during labor.

**Table 4. Women with vitamin D insufficiency, hyperparathyroidism, and PTH below the lower limit.**

| | Cases | | Controls | | |
|---|---|---|---|---|---|
| Variable | n/N | % | n/N | % | *P*-value |
| Vitamin D insufficiency; <50 nmol/l | 16/30 | 53.3% | 10/30 | 33.3% | 0.12 |
| PTH < 1.6 pmol/l | 7/30 | 23.3% | 5/30 | 16.7% | 0.53 |
| PTH > 6.9 pmol/l | 0/30 | 0.0% | 0/30 | 0.0% | NA |

Known risk factors for dystocia include nulliparity, induction of labor, short maternal height, high maternal age, post term gestation and high birthweight [23–25]. In our study, we included nulliparous women and adjusted in secondary analysis for potential risk factors. However, we cannot rule out the possibility that other causal factors not taken into account could bias our results towards an association.

We found no secondary hyperparathyroidism despite 43% of the women being vitamin D insufficient. On the contrary, we found that 20% of the women had a P-PTH under 1.6pmol/l. A possible explanation for this could be a suppression of the maternal PTH by the parathyroid hormone-related peptide (PTHrP). PTHrP has been shown to be released from the cytotrophoblasts and amnion cells [26–28]. PTHrP is present in fetal and gestational tissue where it plays an important role by stimulating the maternal-fetal transfer of calcium through a maternofetal $Ca^{2+}$ gradient in the placenta [28–30]. Another explanation could be the increased synthesis of $1.25(OH)_2D$ during pregnancy, which also might suppress maternal PTH-production [31]. In further studies it could be of interest also to measure levels of $1.25(OH)_2D$ in order to assess whether this active metabolite of vitamin D is of importance to the risk of dystocia.

To our knowledge, no previous study has tested the association between the S-25OHD level and the risk of dystocia. Two studies addressing the association between cesarean section and vitamin D insufficiency showed conflicting results, and both were hampered by limitations. A study on a Pakistani population found no association between the low level of S-25OHD and the risk of having an acute cesarean section [32]. However, the indication for the cesarean section was cephalo-pelvic disproportion and not dystocia. Furthermore, women included in the study were characterized by being malnourished in general. A study from Boston demonstrated that cases who had a primary cesarean section had a lower level of S-25OHD compared to the controls who delivered vaginally (S-25OHD median of 45 nmol/l vs. 63 nmol/l, p = 0.007) [33]. However, also in this study the indication for cesarean section differed from our study, given that only 17 of the 43 cases had dystocia, while the remainder had a primary cesarean as a result of non-reassuring fetal tracing (11 of 43), mal-presentation (6 of 43), and other indications (9 of 43) [33].

The external validity of this study should be restricted to geographic areas with sunlight exposure, clothing and dietary habits, and oral vitamin D supplementation comparable to those in Denmark. Despite the fact that this study displays a significant association between vitamin D and the risk of having an acute cesarean section due to dystocia, we cannot rule out the possibility that our results might be due to other causes or residual confounding. The fact that more than 53% of the cases were vitamin D insufficient supports our hypothesis although more studies are needed to fully elucidate the effect of vitamin D on the risk of dystocia.

## Conclusion

This case-control study showed a significant association between the S-25OHD and the risk of acute cesarean section due to dystocia. Furthermore, more than 40% of the included women were vitamin D insufficient. In view of our results, evaluating the dietary pattern would be important in order to outline the vitamin D insufficiency among pregnant women. Furthermore, the Danish recommendation concerning vitamin D supplementation during pregnancy may need to be reevaluated. However, more studies are needed in order to determine the optimal level of vitamin D supplements for pregnant women in order to lower the prevalence of vitamin D insufficiency and potentially reduce the risk of dystocia.

## Acknowledgments

We would like to thank Professor Erik Thorlund Parner from Department of Public Health–Department of Biostatistics, Aarhus University, for consulting with us on statistics. Furthermore, thank you to Hai Qing for collecting and organizing our blood samples.

## Author Contributions

**Conceptualization:** Lone Hvidman, Lars Rejnmark.

**Data curation:** Christine Rohr Thomsen, Lone Hvidman, Mohammed Rohi Khalil, Niels Uldbjerg.

**Formal analysis:** Christine Rohr Thomsen, Ioanna Milidou, Lars Rejnmark, Niels Uldbjerg.

**Funding acquisition:** Christine Rohr Thomsen, Lars Rejnmark, Niels Uldbjerg.

**Investigation:** Christine Rohr Thomsen, Ioanna Milidou, Lone Hvidman, Mohammed Rohi Khalil, Niels Uldbjerg.

**Methodology:** Christine Rohr Thomsen, Ioanna Milidou, Lone Hvidman, Lars Rejnmark, Niels Uldbjerg.

**Project administration:** Christine Rohr Thomsen, Mohammed Rohi Khalil, Lars Rejnmark, Niels Uldbjerg.

**Resources:** Christine Rohr Thomsen, Ioanna Milidou, Lone Hvidman, Mohammed Rohi Khalil, Lars Rejnmark, Niels Uldbjerg.

**Software:** Christine Rohr Thomsen, Ioanna Milidou.

**Supervision:** Ioanna Milidou, Lone Hvidman, Lars Rejnmark, Niels Uldbjerg.

**Validation:** Ioanna Milidou, Lone Hvidman, Lars Rejnmark, Niels Uldbjerg.

**Visualization:** Christine Rohr Thomsen, Lone Hvidman, Lars Rejnmark, Niels Uldbjerg.

**Writing – original draft:** Christine Rohr Thomsen, Niels Uldbjerg.

**Writing – review & editing:** Christine Rohr Thomsen, Ioanna Milidou, Lone Hvidman, Mohammed Rohi Khalil, Lars Rejnmark, Niels Uldbjerg.

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
