## [Decision Letter · Decision Letter 0]

23 Jun 2020

PONE-D-20-08882

Vitamin D and the risk of dystocia: A case-control study

PLOS ONE

Dear Authors,

Thank you for submitting your manuscript to PLOS ONE. After careful consideration, we feel that it has merit but does not fully meet PLOS ONE’s publication criteria as it currently stands. Therefore, we invite you to submit a revised version of the manuscript that addresses the points raised during the review process.

We look forward to receiving your revised manuscript.

Kind regards,

Salvatore Andrea Mastrolia, M.D.

Academic Editor

PLOS ONE

Journal Requirements:

Reviewers' comments:

Reviewer's Responses to Questions

**Comments to the Author**

1. Is the manuscript technically sound, and do the data support the conclusions?

Reviewer #1: Yes

Reviewer #2: Yes

2. Has the statistical analysis been performed appropriately and rigorously? 

Reviewer #1: Yes

Reviewer #2: Yes

3. Have the authors made all data underlying the findings in their manuscript fully available?

Reviewer #1: Yes

Reviewer #2: Yes

4. Is the manuscript presented in an intelligible fashion and written in standard English?

Reviewer #1: Yes

Reviewer #2: Yes

5. Review Comments to the Author

Reviewer #1: In this study the authors compared the serum levels of 25-OH vitamin D in a group of women delivered by cesarean section for dystocia with a control group of women who delivered vaginally. The experimental question is interesting and relevant, especially considering the increases in the cesarean rate for dystocia over recent decades. This should be considered a preliminary exploration of the issue, which may prove to be quite complex. But it raises an intriguing hypothesis concerning the role of vitamin D in the myometrium. For that reason I think it deserves publication.

There are several issues that I think should be addressed:

1. The abstract loses some focus when it shifts from consideration of the possible relation between vitamin D levels and the risk of dystocia to the need for more vitamin D supplementation (and the implication it might reduce the risk of dystocia). The latter is pure speculation not related to the study findings.

2. The definitions of dystocia used in the study are different from those of others. I realize this is a currently controversial area, but, for example, according to the traditional criteria of Friedman and also Rosen, a woman whose cervix is 4cm dilated is very often still in the latent phase of labor and should not be diagnosed with dystocia. Therefore, some of the study cases delivered for dystocia would have been in normal labor based on some commonly used definitions. Similarly, a woman in the expulsive phase with no descent for one hour has an arrest of descent. I think the manuscript should remind the reader that their definitions of dystocia (and of fetopelvic disproportion) are parochial and may not conform to those used in other geographic areas.

3. Why were patients with suspected disproportion/obstruction excluded? Was this to have a group with dystocia presumably due to poor contractility? If so, this should be explicitly stated. How and by whom was that distinction made?

4. Subjects were taking oral vitamin D. Is there a relationship between the oral dose and serum levels? If so, were the blood draws done at a uniform time in relation to ingestion?

5. Do the authors have 1,25(OH)2 levels from their subjects? If so, these might be helpful. Could changes in the active 1,25 metabolite (which reaches maximal levels at term) reflect myometrial function better than the more long-term stable 25-OH levels?

Reviewer #2: The paper “Vitamin D and the risk of dystocia: A case-control study” aims to evaluate the

association between the vitamin D serum level at labor and the risk of dystocia.

The authors included primiparous women with spontaneous onset of labor and cephalic

presentation of the fetus, who gave birth by cesarean section due to dystocia

(cases) and primiparous women with spontaenous vaginal birth (controls). They collected blood

samples to assess the S-25OHD and the plasma parathyroid hormone (P-PTH) concentrations

among the two groups of patients.

The conclusion was that in the group of cases decreased vitamin D levels were found resulting in a

significant association between the S-25OHD concentrations and the risk of dystocia.

The content of this study is interesting; it is relevant to investigate whether there is a significant association between decresed vitamin D levels and risk of dystocia in order to find out if dystocia during labour could be prevented or predected.

The paper is - regarding the topic and cited references - well written.

However, I have some comments:

Introduction

Line 62: “homogeneous in terms of socioeconomic and nutritional status” specify how the women nutritional status was assessed (food frequency questionnaire? nutrional questionnaire? Food diary?..”)

Material and methods

Table 1 The descending phase; specify the different criteria of distocya with or without epidural analgesia

Table 1 Kjærgaard et al. define as distocya criteria during the active phase of labor a “ Less than 0.5 cm dilatation of cervix/hour, assessed over four hours”. Why did you consider “3-4 hours” as diagnostic criteria?

Exclusion criteria: Was the use of epidural analgesia during labour considered as an exclusion criteria? And the use of drugs affecting vitamin D metabolism?

Line 107: use the abbreviations indicated, “P-PTH” for plasma parathyroid hormone rather than “PTH”

Conclusion

Line 205 “..more than 40% of the included women were vitamin D insufficient. In view of our results, the Danish recommendation concerning vitamin D supplementation during pregnancy may need to be reevaluated”. Supplementation in pregnancy is important to prevent most of the adverse pregancy outcomes but an adequate evaluation of dietary pattern is the first step in order to find out nutritional deficiencies.

References

Line 296 there is a lack of information on publication

Line 299 there is a lack of information on publication

Line 301 there is a lack of information on publication

6. PLOS authors have the option to publish the peer review history of their article (what does this mean?). If published, this will include your full peer review and any attached files.

Reviewer #1: No

Reviewer #2: No

---

## [Author Response · Author response to Decision Letter 0]

24 Aug 2020

Response to Reviewers

Reviewers' Comments to Author:

Reviewer: 1

In this study the authors compared the serum levels of 25-OH vitamin D in a group of women delivered by cesarean section for dystocia with a control group of women who delivered vaginally. The experimental question is interesting and relevant, especially considering the increases in the cesarean rate for dystocia over recent decades. This should be considered a preliminary exploration of the issue, which may prove to be quite complex. But it raises an intriguing hypothesis concerning the role of vitamin D in the myometrium. For that reason I think it deserves publication.

There are several issues that I think should be addressed:

1. The abstract loses some focus when it shifts from consideration of the possible relation between vitamin D levels and the risk of dystocia to the need for more vitamin D supplementation (and the implication it might reduce the risk of dystocia). The latter is pure speculation not related to the study findings.

Thank you for this comment. 

We have deleted the sentence from the abstract in accordance with your suggestion. 

2. The definitions of dystocia used in the study are different from those of others. I realize this is a currently controversial area, but, for example, according to the traditional criteria of Friedman and also Rosen, a woman whose cervix is 4cm dilated is very often still in the latent phase of labor and should not be diagnosed with dystocia. Therefore, some of the study cases delivered for dystocia would have been in normal labor based on some commonly used definitions. Similarly, a woman in the expulsive phase with no descent for one hour has an arrest of descent. I think the manuscript should remind the reader that their definitions of dystocia (and of fetopelvic disproportion) are parochial and may not conform to those used in other geographic areas.

Thank you for making us aware of this source of confusion.

We rephrased in the manuscript in order to avoid confusion. It now reads as follows (Material and Methods; second paragraph);

“Dystocia was defined in accordance with the local guideline (Table 1).”

In addition, we added a comment to the discussion section (Discussion; second paragraph);

“However, the definition of dystocia is based on the local guideline, i.e. other definitions of dystocia could also have been used.”

3. Why were patients with suspected disproportion/obstruction excluded? Was this to have a group with dystocia presumably due to poor contractility? If so, this should be explicitly stated. How and by whom was that distinction made?

Thank you for this comment.

We see, that this was unclear in the manuscript, and we have added a comment about this (Materials and Methods; Study population, third paragraph);

“In order to study women with dystocia because of reduced contractility of the myometrium, we excluded women with cephalo-pelvic disproportion/obstruction; obstetricians with more than 15 years of clinical experience made this distinction (NU, LH & MK).“

4. Subjects were taking oral vitamin D. Is there a relationship between the oral dose and serum levels? If so, were the blood draws done at a uniform time in relation to ingestion?

Blood samples were obtained between 9 am and noon (12 am) in order to avoid any influence by the potential circadian rhythm of PTH. Of importance, we measured vitamin status in terms of 25-OHD levels, whereas vitamin D supplements are provided as cholecalciferol (vitamin D3). As the serum half-life for S-25OHD is about 3 to 5 weeks, we expect that the serum levels would be minimally influenced by difference in pattern (time of day) regarding ingestion of vitamin D supplements. 

The manuscript has now been clarified by specifically stating S-25OHD when referring to serum levels (instead of stating “vitamin D levels”) 

To further investigate the relationship between the oral dose and serum levels, we have calculated a Spearman’s oh, and added to the manuscript (Material and Methods; statistical analysis; first paragraph);

“Furthermore, we calculated correlation between studied variables by calculation Spearman's oh (ρ).”

In addition, we added to the result section (Results; fourth paragraph); 

“Serum 25-OHD levels correlated significantly with daily dose of vitamin D from supplements (ρ 0.264; p=0.04).”

5. Do the authors have 1,25(OH)2 levels from their subjects? If so, these might be helpful. Could changes in the active 1,25 metabolite (which reaches maximal levels at term) reflect myometrial function better than the more long-term stable 25-OH levels?

An interesting aspect! Unfortunately, we do not have the 1,25(OH)2 levels from the included women, though this would have been of interest in regard to the final result. In the revised manuscript, we have now added this as a suggestion for further studies (Discussion; fourth paragraph); 

“In further studies it could be of interest also to measure levels of 1.25(OH)2D in order to assess whether this active metabolite of vitamin D is of importance to the risk of dystocia.”

Reviewer #2: The paper “Vitamin D and the risk of dystocia: A case-control study” aims to evaluate the

association between the vitamin D serum level at labor and the risk of dystocia.

The authors included primiparous women with spontaneous onset of labor and cephalic

presentation of the fetus, who gave birth by cesarean section due to dystocia

(cases) and primiparous women with spontaenous vaginal birth (controls). They collected blood samples to assess the S-25OHD and the plasma parathyroid hormone (P-PTH) concentrations among the two groups of patients. The conclusion was that in the group of cases decreased vitamin D levels were found resulting in a significant association between the S-25OHD concentrations and the risk of dystocia.

The content of this study is interesting; it is relevant to investigate whether there is a significant association between decresed vitamin D levels and risk of dystocia in order to find out if dystocia during labour could be prevented or predected.

The paper is - regarding the topic and cited references - well written.

However, I have some comments:

Introduction

Line 62: “homogeneous in terms of socioeconomic and nutritional status” specify how the women nutritional status was assessed (food frequency questionnaire? nutrional questionnaire? Food diary?..”)

Thank you for this comment. The statement about nutritional status is more a general consideration regarding the Danish population and do not reflect any specific data. As we do not have any reference to include, we will remove this statement if the Academic Editor states this will be the most appropriate. 

Material and methods

Table 1 The descending phase; specify the different criteria of distocya with or without epidural analgesia

Thanks for making us aware of this lack of information. 

Today, most definitions of dystocia include the use of epidural analgesia, however, when this study was initiated this was not the case. This is a limitation to the study, and we have added to the discussion section (Discussion; second paragraph); 

“Furthermore, in accordance with the local guideline the use of epidural analgesia was not addressed in the definition of dystocia, which might be considered as a limitation to the study.”

In accordance with your comment, we have reviewed our study population in order to outline the use of epidural analgesia during labor. We have performed a linear regression adjusting for the use of epidural analgesia during labor, which did not change the result. We have added to the manuscript (Material and Method; fourth paragraph)

“…and the use of epidural analgesia during labor…”

(Material and Method; Statistical analysis; second paragraph)

“…and the use of epidural analgesia during labor…”

(Results; table 2)

Use of epidural analgesia during labor n (%) (chi squared) 13(43.3%) NA 3(10%) NA 0.004

(Results; second paragraph)

“…and the use of epidural analgesia during labor (P=0.02)…”

(Discussion; second paragraph)

“However, our results did not change when adjusting for the use of epidural analgesia during labor.”

Table 1 Kjærgaard et al. define as distocya criteria during the active phase of labor a “ Less than 0.5 cm dilatation of cervix/hour, assessed over four hours”. Why did you consider “3-4 hours” as diagnostic criteria?

Thanks for this comment. The guideline used in this study did not adhere strictly to the guideline presented by Kjærgaard et al, but were modified in accordance with the local guideline. 

Reviewer 1 also addressed the fact, that a local guideline was used in the study. We rephrased in the manuscript in order to avoid confusion. It now reads as follows (Material and Methods; second paragraph);

“Dystocia was defined in accordance with the local guideline (Table 1).”

In addition, we added a comment to the discussion section (Discussion; second paragraph);

“However, the definition of dystocia is based on the local guideline, i.e. other definitions of dystocia could also have been used.”

Exclusion criteria: Was the use of epidural analgesia during labour considered as an exclusion criteria? And the use of drugs affecting vitamin D metabolism?

The use of epidural analgesia was not addressed in the definition of dystocia, which is a limitation to the study. This limitation have been added to the manuscript (Discussion; second paragraph).

 “Furthermore, in accordance with the local guideline the use of epidural analgesia was not addressed in the definition of dystocia, which might be considered as a limitation to the study.”

In accordance with your comment, we have reviewed our study population and found that no included women took any drug that potentially could affect the vitamin D metabolism or the birth process. We have specified this in the manuscript (Results; first paragraph).

“None of the included women took any drugs affecting the vitamin D metabolism or the birth 

process.”

Line 107: use the abbreviations indicated, “P-PTH” for plasma parathyroid hormone rather than “PTH”

Thank you for this comment. We have changed the wording in accordance with the suggestions. 

Conclusion

Line 205 “..more than 40% of the included women were vitamin D insufficient. In view of our results, the Danish recommendation concerning vitamin D supplementation and during pregnancy may need to be reevaluated”. Supplementation in pregnancy is important to prevent most of the adverse pregancy outcomes but an adequate evaluation of dietary pattern is the first step in order to find out nutritional deficiencies.

In interesting aspect! In accordance with our suggesting, we have added to the manuscript (Conclusion).

“In view of our results, evaluating the dietary pattern would be important in order to outline the vitamin D insufficiency among pregnant women. Furthermore, the Danish recommendation concerning vitamin D supplementation during pregnancy may need to be reevaluated.”

References

Line 296 there is a lack of information on publication

Line 299 there is a lack of information on publication

Line 301 there is a lack of information on publication

Thank you for making us aware of this lack of information. We have now updated the reference list. 

6. PLOS authors have the option to publish the peer review history of their article (what does this mean?). If published, this will include your full peer review and any attached files.

Do you want your identity to be public for this peer review? For information about this choice, including consent withdrawal, please see our Privacy Policy.

Reviewer #1: No

Reviewer #2: No

While revising your submission, please upload your figure files to the Preflight Analysis and Conversion Engine (PACE) digital diagnostic tool, https://pacev2.apexcovantage.com/. PACE helps ensure that figures meet PLOS requirements. To use PACE, you must first register as a user. Registration is free. Then, login and navigate to the UPLOAD tab, where you will find detailed instructions on how to use the tool. If you encounter any issues or have any questions when using PACE, please email PLOS at figures@plos.org. Please note that Supporting Information files do not need this step

---

## [Decision Letter · Decision Letter 1]

28 Sep 2020

Vitamin D and the risk of dystocia: A case-control study

PONE-D-20-08882R1

Dear Authors,

We’re pleased to inform you that your manuscript has been judged scientifically suitable for publication and will be formally accepted for publication once it meets all outstanding technical requirements.

Kind regards,

Salvatore Andrea Mastrolia, M.D.

Academic Editor

PLOS ONE

Reviewers' comments:

Reviewer's Responses to Questions

**Comments to the Author**

1. If the authors have adequately addressed your comments raised in a previous round of review and you feel that this manuscript is now acceptable for publication, you may indicate that here to bypass the “Comments to the Author” section, enter your conflict of interest statement in the “Confidential to Editor” section, and submit your "Accept" recommendation.

Reviewer #1: All comments have been addressed

Reviewer #2: All comments have been addressed

2. Is the manuscript technically sound, and do the data support the conclusions?

Reviewer #1: Yes

Reviewer #2: Yes

3. Has the statistical analysis been performed appropriately and rigorously? 

Reviewer #1: Yes

Reviewer #2: Yes

4. Have the authors made all data underlying the findings in their manuscript fully available?

Reviewer #1: Yes

Reviewer #2: Yes

5. Is the manuscript presented in an intelligible fashion and written in standard English?

Reviewer #1: Yes

Reviewer #2: Yes

6. Review Comments to the Author

Reviewer #1: (No Response)

Reviewer #2: (No Response)

7. PLOS authors have the option to publish the peer review history of their article (what does this mean?). If published, this will include your full peer review and any attached files.

Reviewer #1: No

Reviewer #2: No

---

## [Editor Report · Acceptance letter]

5 Oct 2020

PONE-D-20-08882R1 

Vitamin D and the risk of dystocia: A case-control study 

Dear Dr. Thomsen:

I'm pleased to inform you that your manuscript has been deemed suitable for publication in PLOS ONE. Congratulations! Your manuscript is now with our production department. 

Kind regards, 

on behalf of

Dr. Salvatore Andrea Mastrolia 

Academic Editor

PLOS ONE